# ATTENTION-DRIVEN ROBOTIC MANIPULATION

## ABSTRACT

Despite the success of reinforcement learning methods, they have yet to have their breakthrough moment when applied to a broad range of robotic manipulation tasks. This is partly due to the fact that reinforcement learning algorithms are notoriously difficult and time consuming to train, which is exacerbated when training from images rather than full-state inputs. As humans perform manipulation tasks, our eyes closely monitor every step of the process with our gaze focusing sequentially on the objects being manipulated. With this in mind, we present our Attention-driven Robotic Manipulation (ARM) algorithm, which is a general manipulation algorithm that can be applied to a range of real-world sparse-rewarded tasks without any prior task knowledge. ARM splits the complex task of manipulation into a 3 stage pipeline: (1) a Q-attention agent extracts interesting pixel locations from RGB and point cloud inputs, (2) a next-best pose agent that accepts crops from the Q-attention agent and outputs poses, and (3) a control agent that takes the goal pose and outputs joint actions. We show that current state-of-the-art reinforcement learning algorithms catastrophically fail on a range of RLBench tasks, whilst ARM is successful within a few hours.

## 1 INTRODUCTION

Despite their potential, continuous-control reinforcement learning (RL) algorithms have many flaws: they are notoriously data hungry, often fail with sparse rewards, and struggle with long-horizon tasks. The algorithms for both discrete and continuous RL are almost always evaluated on benchmarks that give shaped rewards (Brockman et al., 2016; Tassa et al., 2018), a privilege that is not feasible for training real-world robotic application across a broad range of tasks. Motivated by the observation that humans focus their gaze close to objects being manipulated (Land et al., 1999), we propose an Attention-driven Robotic Manipulation (ARM) algorithm that consists of a series of algorithm-agnostic components, that when combined, results in a method that is able to perform a range of challenging, sparsely-rewarded manipulation tasks.

Our algorithm operates through a pipeline of modules: our novel **Q-attention** module first extracts interesting pixel locations from RGB and point cloud inputs by treating images as an environment, and pixel locations as actions. Using the pixel locations we crop the RGB and point cloud inputs, significantly reducing input size, and feed this to a next-best-pose continuous-control agent that outputs 6D poses, which is trained with our novel **confidence-aware critic**. These goal poses are then used by a control algorithm that continuously outputs motor velocities.

As is common with sparsely-rewarded tasks, we improve initial exploration through the use of demonstrations. However, rather than simply inserting these directly into the replay buffer, we use a **keyframe discovery** strategy that chooses interesting keyframes along demonstration trajectories that is fundamental to training our Q-attention module. Rather than storing the transition from an initial state to a keyframe state, we use our **demo augmentation** method which also stores the transition from intermediate points along a trajectories to the keyframe states; thus greatly increasing the proportion of initial demo transitions in the replay buffer.

All of these improvements result in an algorithm that starkly outperforms other state-of-the-art methods when evaluated on 10 RLBench (James et al., 2020) tasks (Figure 1) that range in difficulty. To summarise, we propose the following contributions: **(1)** An off-policy hard attention mechanism that is learned via Q-Learning, rather than the on-policy hard attention and soft attention that is commonly seen in the NLP and vision community; **(2)** A confidence-aware Q function that predicts

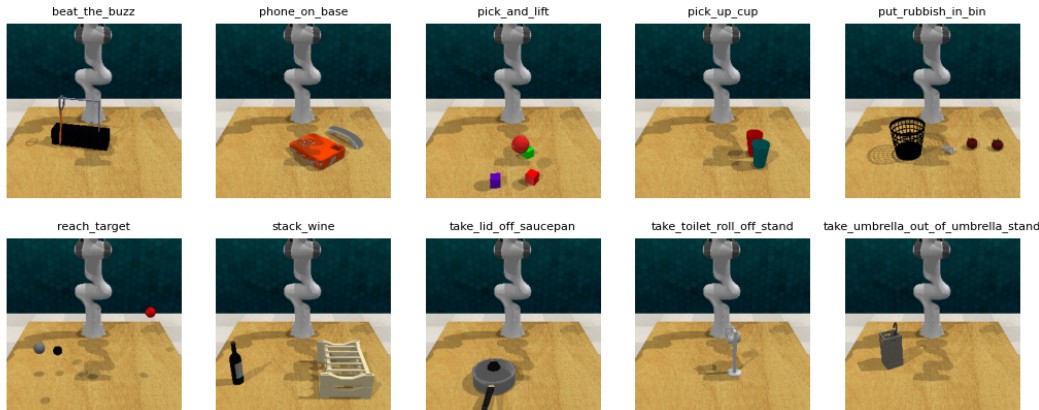

Figure 1: The 10 RLBench tasks used for evaluation. Current state-of-the-art reinforcement learning algorithms catastrophically fail on all tasks, whilst our method succeeds within a modest number of steps. Note that the positions of objects are placed randomly at the beginning of each episode.

pixel-wise Q values and confidence values, resulting in improved convergence times; **(3)** A keyframe discovery strategy and demo augmentation method that go hand-in-hand to improve the utilisation of demonstrations in RL.

## 2 RELATED WORK

The use of reinforcement learning (RL) is prevalent in many areas of robotics, including legged robots (Kohl & Stone, 2004; Hwangbo et al., 2019), aerial vehicles (Sadeghi & Levine, 2017), and manipulation tasks, such as pushing (Finn & Levine, 2017), peg insertion (Levine et al., 2016; Zeng et al., 2018; Lee et al., 2019), throwing (Ghadirzadeh et al., 2017; Zeng et al., 2020), ball-in-cup (Kober & Peters, 2009), cloth manipulation (Matas et al., 2018), and grasping (Kalashnikov et al., 2018; James et al., 2019b). Despite the abundance of work in this area, there has yet to be a general manipulation method that can tackle a range of challenging, sparsely-rewarded tasks without needing access to privileged simulation-only abilities (e.g. reset to demonstrations (Nair et al., 2018), asymmetric actor-critic (Pinto et al., 2018), reward shaping (Rajeswaran et al., 2018), and auxiliary tasks (James et al., 2017)).

Crucial to our method is the proposed Q-attention. Soft and hard attention are prominent methods in both natural language processing (NLP) (Bahdanau et al., 2015; Vaswani et al., 2017; Devlin et al., 2018) and computer vision (Xu et al., 2015; Zhang et al., 2019). Soft attention deterministically multiplies an attention map over the image feature map, whilst hard attention uses the attention map stochastically to sample one or a few features on the feature map (which is optimised by maximising an approximate variational lower bound or equivalently via (on-policy) REINFORCE (Williams, 1992)). Given that we perform non-differentiable cropping, our Q-attention is closest to hard attention, but with the distinction that we learn this in an off-policy way. This is key, as 'traditional' hard attention is unable to be used in an off-policy setting. We therefore see Q-attention as an off-policy hard attention. We elaborate further on these differences in Section 4.1.

Our proposed confidence-aware critic (used to train the next-best pose agent) takes its inspiration from the pose estimation community (Wang et al., 2019; Wada et al., 2020). There exists a small amount of work in estimating uncertainty with Q-learning in discrete domains (Clements et al., 2019; Hoel et al., 2020); our work uses a continuous Q-function to predict both Q and confidence values for each pixel, which lead to improved stability when training, and is not used during action selection.

Our approach makes use of demonstrations, which has been applied in a number of works (Vecerik et al., 2017; Matas et al., 2018; Kalashnikov et al., 2018; Nair et al., 2018), but while successful, they make limited use of the demonstrations and still can take many samples to converge. Rather than simply inserting these directly into the replay buffer, we instead make sure of our keyframe discovery and demo augmentation to maximise demonstration utility.

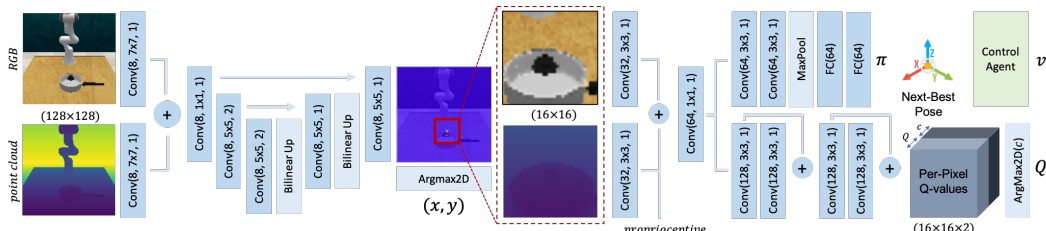

Figure 2: Summary and architecture of our method. RGB and organised point cloud crops are made by extracting pixel locations from our Q-attention module. These crops are then fed to a continuous control RL algorithm that suggests next-best poses that is trained with a confidence-aware critic. The next best pose is given to a goal-condition control agent that outputs joint velocities. Conv block represented as *Conv(#channels, filter size, strides)*.

## 3 BACKGROUND

The reinforcement learning paradigm assumes an agent interacting with an environment consisting of states $\mathbf{s} \in \mathcal{S}$, actions $\mathbf{a} \in \mathcal{A}$, and a reward function $R(\mathbf{s}_t, \mathbf{a}_t)$, where $\mathbf{s}_t$ and $\mathbf{a}_t$ are the state and action at time step $t$ respectively. The goal of the agent is then to discover a policy $\pi$ that results in maximising the expectation of the sum of discounted rewards: $\mathbb{E}_\pi[\sum_t \gamma^t R(\mathbf{s}_t, \mathbf{a}_t)]$, where future rewards are weighted with respect to the discount factor $\gamma \in [0, 1)$. Each policy $\pi$ has a corresponding value function $Q(s, a)$, which represents the expected return when following the policy after taking action $\mathbf{a}$ in state $\mathbf{s}$.

Our Q-attention module builds from Deep Q-learning (Mnih et al., 2015), a method that approximated the value function $Q_\theta$, with a deep convolutional network, whose parameters $\theta$ are optimised by sampling mini-batches from a replay buffer $\mathcal{D}$ and using stochastic gradient descent to minimise the loss: $\mathbb{E}_{(\mathbf{s}_t, \mathbf{a}_t, \mathbf{s}_{t+1}) \sim \mathcal{D}}[(\mathbf{r} + \gamma max_{\mathbf{a}'} Q_{\theta'}(\mathbf{s}_{t+1}, \mathbf{a}') - Q_\theta(\mathbf{s}_t, \mathbf{a}_t))^2]$, where $Q_{\theta'}$ is a target network; a periodic copy of the online network $Q_\theta$ which is not directly optimised. Our next-best pose agent builds upon SAC (Haarnoja et al., 2018), however, the agent is compatible with any off-policy, continuous-control RL algorithm. SAC is a maximum entropy RL algorithm that, in addition to maximising the sum of rewards, also maximises the entropy of a policy: $\mathbb{E}_\pi[\sum_t \gamma^t [R(\mathbf{s}_t, \mathbf{a}_t) + \alpha \mathcal{H}(\pi(\cdot|\mathbf{s}_t))]]$, where $\alpha$ is a temperature parameter that determines the relative importance between the entropy and reward. The goal then becomes to maximise a soft Q-function by minimising the following Bellman residual:

$$J_Q(\theta) = \mathbb{E}_{(\mathbf{s}_t, \mathbf{a}_t, \mathbf{s}_{t+1}) \sim \mathcal{D}}[((\mathbf{r} + \gamma Q_{\theta'}(\mathbf{s}_{t+1}, \pi_\phi(\mathbf{s}_{t+1})) - \alpha \log \pi_\phi(\mathbf{a}_t|\mathbf{s}_t)) - Q_\theta(\mathbf{s}_t, \mathbf{a}_t))^2]. \quad (1)$$

The policy is updated towards the Boltzmann policy with temperature $\alpha$, with the Q-function taking the role of (negative) energy. Specifically, the goal is to minimise the Kullback-Leibler divergence between the policy and the Boltzman policy:

$$\pi_{\text{new}} = \arg \min_{\pi' \in \Pi} D_{\text{KL}} \left( \pi'(\cdot|s_t) \, \middle\| \, \frac{\frac{1}{\alpha} \exp(Q^{\pi_{\text{old}}}(s_t, \cdot))}{Z^{\pi_{\text{old}}}(s_t)} \right). \quad (2)$$

Minimising the expected KL-divergence to learn the policy parameters was shown to be equivalent to maximising the expected value of the soft Qfunction:

$$J_\pi(\phi) = \mathbb{E}_{\mathbf{s}_t \sim \mathcal{D}} \left[ \mathbb{E}_{\mathbf{a} \sim \pi_\phi} [\alpha \log(\pi_\phi(\mathbf{a}_t|\mathbf{s}_t)) - Q_\rho^\pi(\mathbf{s}_t, \mathbf{a}_t)] \right]. \quad (3)$$

## 4 METHOD

Our method can be split into a 3-phase pipeline. Phase 1 (Section 4.1) consists of a high-level pixel agent that selects areas of interest using our novel Q-attention module. Phase 2 (Section 4.2) consists

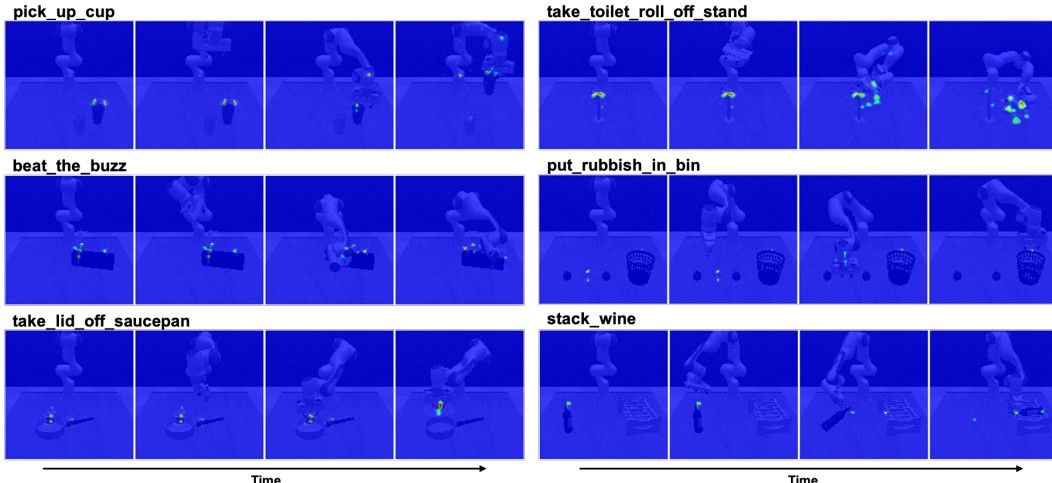

Figure 3: Visualising the Q values across 4 different points in time on 6 tasks. At each step, RGB and organised point cloud crops are made by extracting pixel locations that have the highest Q-value. We can see that as time progresses, the attention strength shifts depending on progress in the task; e.g. *'stack_wine'* starts with high attention on the bottle, but after grasping, attention shifts to the wine rack.

of a next-best pose prediction phase where the pixel location from the previous phase is used to crop the incoming observations and then predict a 6D pose. Finally, phase 3 (Section 4.3) is a low-level control agent that accepts the predicted next-best pose and executes a series of actions to reach the given goal. Before training, we fill the replay buffer with demonstrations using our keyframe discovery and demo augmentation strategy (Section 4.4) that significantly improves training speed. The full pipeline is summarised in Figure 2 and Algorithm 1.

All experiments are run in RLBench (James et al., 2020), a large-scale benchmark and learning environment for vision-guided manipulation built around CoppeliaSim (Rohmer et al., 2013) and PyRep (James et al., 2019a). At each time step, we extract an observation from the front-facing camera that consists of an RGB image $\mathbf{b}$ and a depth image $\mathbf{d}$, along with proprioceptive information $\mathbf{z}$ from the arm (consisting of end-effector pose and gripper open/close state). Using known camera intrinsics and extrinsics, we process each depth image to produce a point cloud $\mathbf{p}$ (in world coordinates) projected from the view of the front-facing camera, producing a $(H \times W \times 3)$ 'image'.

## 4.1 Q-ATTENTION

Motivated by the role of vision and eye movement in the control of human activities (Land et al., 1999), we propose a Q-attention module that, given RGB and organised point cloud inputs, outputs 2D pixel locations of the next area of interest. With these pixel locations, we crop the RGB and organised point cloud inputs and thus drastically reduce the input size to the next stage of the pipeline. Our Q-attention is explicitly learned via Q-learning, where images are treated as the 'environment', and pixel locations are treated as the 'actions'.

Given our Q-attention function $QA_\theta$, we extract the coordinates of pixels with the highest value:

$$(\mathbf{x}_t, \mathbf{y}_t) = \operatorname*{argmax}_{\mathbf{a}'} 2D \, QA_\theta(\mathbf{s}_t, \mathbf{a}'). \tag{4}$$

The parameters of the Q-attention are optimised by using stochastic gradient descent to minimise the loss:

$$J_{QA}(\theta) = \mathop{\mathbb{E}}_{(\mathbf{s}_t, \mathbf{a}_t, \mathbf{s}_{t+1}) \sim \mathcal{D}} [(\mathbf{r} + \gamma \max_{\mathbf{a}'} 2D \, QA_{\theta'}(\mathbf{s}_{t+1}, \mathbf{a}') - QA_\theta(\mathbf{s}_t, \mathbf{a}_t))^2 + \|QA\|], \tag{5}$$

where $\mathbf{s} = (\mathbf{b}, \mathbf{p})$, $QA_{\theta'}$ is the target Q-function, and $\|QA\|$ is an $L2$ loss on the per-pixel output of the Q function (which we call *Q regularisation*); in practice, we found that this leads to increased

robustness against the common problem of overestimation of Q values. The Q-attention network follows a light-weight U-Net style architecture (Ronneberger et al., 2015), which is summarised in Figure 2. Example per-pixel outputs of the Q-attention are shown in Figure 3. With the suggested coordinates from Q-attention, we perform a ($16 \times 16$) crop on both the ($128 \times 128$) RGB and organised point cloud data: $\mathbf{b}', \mathbf{p}' = crop(\mathbf{b}, \mathbf{p}, (\mathbf{x}, \mathbf{y}))$.

Notably, there is no explicit reward for choosing a pixel, but instead an implicit reward that comes from the output of the method pipeline as a whole (i.e. the same reward signal is used to train both the Q-attention and the next-best pose agent). This leads to a cyclic dependency between the two agents: the lower-level next-best pose agent relies on receiving good crops from the Q-attention agent, whilst the Q-attention agent needs the next-best pose agent to perform well in order to get its implicit reward. This is where delicate handling of demonstrations is key, which we discuss in Section 4.4.

The module shares similar human-inspired motivation to the attention seen in NLP (Bahdanau et al., 2015; Vaswani et al., 2017; Devlin et al., 2018) and computer vision (Xu et al., 2015; Zhang et al., 2019), but differs in its formulation. Soft attention multiplies an attention map over the image feature map, whilst hard attention uses the attention map to sample one or a few features on the feature maps or inputs. Given that we perform non-differentiable cropping, we categorise our Q-attention as hard attention, but with 2 core differences: (1) most importantly, 'traditional' hard attention is optimised (on-policy) by maximising an approximate variational lower bound or equivalently via REINFORCE (Williams, 1992), whereas our Q-attention is trained off-policy; this is crucial because our demonstration data, by definition, is off-policy, and therefore renders existing hard attention approaches unusable for demonstration-driven RL. (2) The output of 'traditional' hard attention carry different semantics: a score function in the case of REINFORCE hard attention, and Q-value (expected cumulative reward of choosing that crop) in the case of Q-attention.

## 4.2 NEXT-BEST POSE AGENT

Our next-best pose agent accepts cropped RGB $\mathbf{b}'$ and organised point cloud $\mathbf{p}'$ inputs, and outputs a 6D pose. This next-best pose agent is run every time the robot reaches the previously selected pose. We represent the 6D pose via a translation $\mathbf{e} \in \mathcal{R}^3$ and a unit quaternion $\mathbf{q} \in \mathcal{R}^4$, and restrict the $w$ output of $\mathbf{q}$ to a positive number, therefore restricting the network to output unique unit quaternions. The gripper action $\mathbf{h} \in \mathcal{R}^1$ lies between 0 and 1, which is then discretised to a binary open/close value. The combined action therefore is $\mathbf{a} = \{\mathbf{e}, \mathbf{q}, \mathbf{h}\}$.

To train this next-best pose agent, we use a modified version of SAC (Haarnoja et al., 2018) where we modify the soft Q-function (Equation 1) to be a confidence-aware soft Q-function. Recent work in 6D pose estimation (Wang et al., 2019; Wada et al., 2020) has seen the inclusion of a confidence score $c$ with the pose prediction output for each dense-pixel. Inspired by this, we augment our Q function with a per-pixel confidence $c_{ij}$, where we output a confidence score for each Q-value prediction (resulting in a ($16 \times 16 \times 2$) output). To achieve this, we weight the per-pixel Bellman loss with the per-pixel confidence, and add a confidence regularisation term:

$$J_{Q^\pi}(\rho) = \mathbb{E}_{(\mathbf{s}_t, \mathbf{a}_t, \mathbf{s}_{t+1}) \sim \mathcal{D}} [((\mathbf{r} + \gamma Q_{\rho'}^\pi(\mathbf{s}_{t+1}, \pi_\phi(\mathbf{s}_{t+1})) - \alpha \log \pi(\mathbf{a}_t|\mathbf{s}_t)) - Q_\rho^\pi(\mathbf{s}_t, \mathbf{a}_t))^2 c - w \log(c)], \tag{6}$$

where $\mathbf{s} = (\mathbf{b}', \mathbf{p}', \mathbf{z})$. With this, low confidence will result in a low Bellman error but would incur a high penalty from the second term, and vice versa. We use the Q value that has the highest confidence when training the actor. As an aside, we also tried applying this confidence-aware method to the policy, though empirically we found no improvement. In practice we make use of the clipped double-Q trick (Fujimoto et al., 2018), which takes the minimum Q-value between two Q networks, but have omitted in the equations for brevity. Finally, the actor's policy parameters can be optimised by minimising the loss as defined in Equation 3.

## 4.3 CONTROL AGENT

Given the next-best pose suggestion from the previous stage, we give this to a goal-conditioned control function $f(\mathbf{s}_t, \mathbf{g}_t)$, which given state $\mathbf{s}_t$ and goal $\mathbf{g}_t$, outputs motor velocities that drives the

---

**Algorithm 1** ARM

---

Initialise Q-attention networks $QA_{\theta_1}$, $QA_{\theta_2}$, critic networks $Q_{\rho_1}^\pi$, $Q_{\rho_2}^\pi$, and actor network $\pi_\phi$ with random parameters $\theta_1, \theta_2, \rho_1, \rho_2, \phi$
Initialise target networks $\theta_1' \leftarrow \theta_1, \theta_2' \leftarrow \theta_2, \rho_1' \leftarrow \rho_1, \rho_2' \leftarrow \rho_2$
Initialise replay buffer $\mathcal{D}$ with demos and apply keyframe selection and demo augmentation
**for** each iteration **do**
 **for** each environment step **do**
  $(\mathbf{b}_t, \mathbf{p}_t, \mathbf{z}_t) \leftarrow \mathbf{s}_t$
  $(x_t, y_t) \leftarrow \text{argmax} 2D_{\mathbf{a}'} QA_\theta((\mathbf{b}_t, \mathbf{p}_t), \mathbf{a}')$     ▷ Use Q-attention to get pixel coords
  $\mathbf{b}_t', \mathbf{p}_t' \leftarrow crop(\mathbf{b}_t, \mathbf{p}_t, (x_t, y_t))$
  $\mathbf{a}_t \sim \pi_\phi(\mathbf{a}_t | (\mathbf{b}_t', \mathbf{p}_t', \mathbf{z}_t))$        ▷ Sample pose from the policy
  **while** target not reached **do**
   $v \leftarrow f(\mathbf{s}, \mathbf{a}_t)$          ▷ Get joint velocities from control agent
   $\mathbf{s}_{t+1}, \mathbf{r} \leftarrow env.step(v)$
  $\mathcal{D} \leftarrow \mathcal{D} \cup \{(\mathbf{s}_t, \mathbf{a}_t, \mathbf{r}, \mathbf{s}_{t+1}, (x_t, y_t))\}$    ▷ Store the transition in the replay pool
 **for** each gradient step **do**
  $\theta_i \leftarrow \theta_i - \lambda_{QA} \hat{\nabla}_{\theta_i} J_{QA}(\theta_i)$ for $i \in \{1, 2\}$    ▷ Update Q-attention parameters
  $\rho_i \leftarrow \rho_i - \lambda_{Q^\pi} \hat{\nabla}_{\rho_i} J_{Q^\pi}(\rho_i)$ for $i \in \{1, 2\}$    ▷ Update critic parameters
  $\phi \leftarrow \phi - \lambda_\pi \hat{\nabla}_\phi J_\pi(\phi)$         ▷ Update policy weights
  $\theta_i' \leftarrow \tau \theta_i + (1 - \tau)\theta_i'$ for $i \in \{1, 2\}$   ▷ Update Q-attention target network weights
  $\rho_i' \leftarrow \tau \rho_i + (1 - \tau)\rho_i'$ for $i \in \{1, 2\}$     ▷ Update critic target network weights

---

end-effector towards the goal. This function can take on many forms, but two noteworthy solutions would be either motion planning in combination with a feedback-control or a learnable policy trained with imitation/reinforcement learning. Given that the environmental dynamics are limited in the benchmark, we opted for the motion planning solution.

Given the target pose, we perform path planning using the SBL (Sánchez & Latombe, 2003) planner within OMPL (Şucan et al., 2012), and use Reflexxes Motion Library for on-line trajectory generation. If the target pose is out of reach, we terminate the episode and supply a reward of $-1$. This path planning and trajectory generation is conveniently encapsulated by the '*ABS_EE_POSE_PLAN_WORLD_FRAME*' action mode in RLBench (James et al., 2020).

## 4.4 KEYFRAME SELECTION & DEMO AUGMENTATION

In this section, we outline how we maximise the utility of given demonstrations in order to complete sparsely reward tasks. We assume to have a teacher policy $\pi^*$ (e.g. motion planners or human teleoperatives) that can generate trajectories consisting of a series of states and actions: $\tau = [(\mathbf{s}_1, \mathbf{a}_1), \dots, (\mathbf{s}_T, \mathbf{a}_T)]$. In this case, we assume that the demonstrations come from RLBench (James et al., 2020).

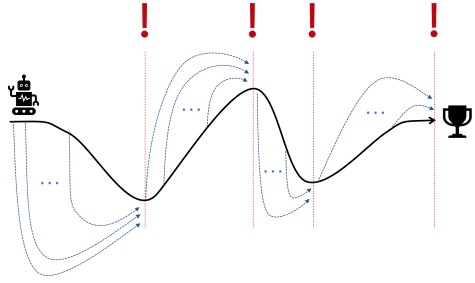

Figure 4: Keyframe selection and demo augmentation, where the black line represents a trajectory, '**!**' represents keyframes, and dashed blue lines represent the augmented transitions to the keyframes.

The **keyframe selection** process iterates over each of the demo trajectories $\tau$ and runs each of the state-action pairs $(\mathbf{s}, \mathbf{a})$ through a function $K : \mathbb{R}^D \rightarrow \mathbb{B}$ which outputs a boolean deciding if the given trajectory point should be treated as a keyframe. The keyframe function $K$ could include a number of constraints. In practice we found that performing a disjunction over two simple conditions worked well; these included (1) change in gripper state (a common occurrence when something is grasped or released), and (2) velocities approaching near zero (a common occurrence when entering pre-grasp poses or entering a new phase of a task). It is likely that as tasks get more complex, $K$ will inevitably need to become more sophisticated via learning

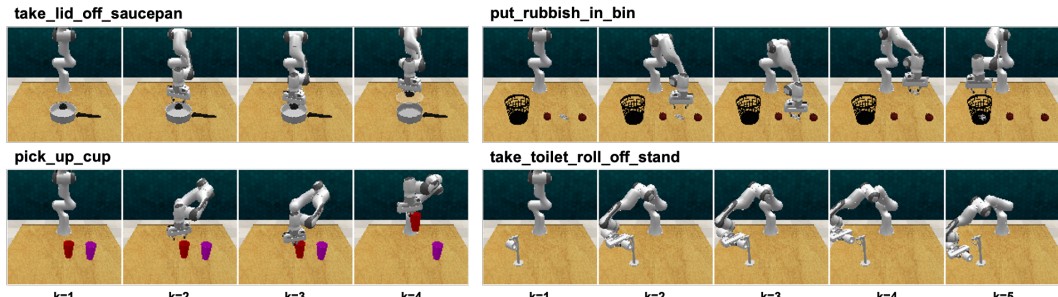

Figure 5: Visualising RGB observations of keyframes from the keyframe selection process on 4 tasks. Here $k$ is the keyframe number.

or simply through more conditions, e.g. sudden changes in direction or joint velocity, large changes in pixel values, etc. Figure 5 shows RGB observations from the keyframe selection process from 4 tasks.

At each keyframe, we use the known camera intrinsics and extrinsics to project the end-effector pose at state $s_{t+1}$ into the image plane of state $s_t$, giving us pixel locations of the end-effector at the next keyframe. This stage is crucial to breaking the cyclic dependency (mentioned in Section 4.1) between the Q-attention and next-best pose agent, as these projected pixel coordinates act as optimal actions for the Q-attention agent.

Using this keyframe selection method, each trajectory results in $N = length(keyframes)$ transitions being stored into the replay buffer. To further increase the utility of demonstrations, we apply **demo augmentation** which stores the transition from an intermediate point along the trajectories to the keyframe states. Formally, for each point $(s_t, a_t)$ along the trajectory starting from keyframe $k_i$, we calculate the transformation of the end-effector pose (taken from $s_t$) at time step $t$ to the end-effector pose at the time step associated with keyframe $k_{i+i}$. This transformation can then be used as the action for the next-best pose agent. We repeat this process for every $M$th point along the trajectory (which we set to $M = 5$). The keyframe selection and demo augmentation is visualised in Figure 4.

## 5 RESULTS

In this section, we aim to answer the following questions: (1) Are we able to successfully learn across a range of sparsely-rewarded manipulation tasks? (2) Which of our proposed components contribute the most to our success? (3) How sensitive is our method to the number of demonstrations and to the crop size? To answer these, we benchmark our approach using RLBench (James et al., 2020). Of the 100 tasks, we select 10 (shown in Figure 1) that we believe to be achievable from using only the front-facing camera. We leave tasks that require multiple cameras to future work. RLBench was chosen due to its emphasis on vision-based manipulation benchmarking and because it gives access to a wide variety of tasks with demonstrations. Each task has a completely sparse reward of $+1$ which is given only on task completion, and $0$ otherwise.

The first of our questions can be answered by attending to Figure 6. All baseline algorithms (SAC, TD3 and QT-Opt) are in their 'vanilla' form, and do not contain any of our proposed contributions: Q-attention, confidence-aware critic, keyframe selection, and demo augmentation. All methods receive the exact same 200 demonstration sequences, which are loaded into the replay buffer prior to training. The baseline agents are architecturally similar to the next-best pose agent, but with a few differences to account for missing Q-attention (and so receives the full, uncropped RGB and organised point cloud data) and missing confidence-aware critic (and so outputs single Q-values rather than per-pixel values). Specifically, the architecture uses the same RGB and point cloud fusion as shown in Figure 2. Feature maps from the shared representation are concatenated with the reshaped proprioceptive input and fed to both the actor and critic. The baseline actor uses 3 convolution layers (64 channels, $3 \times 3$ filter size, 2 stride), who's output feature maps are maxpooled and sent through 2 dense layers (64 nodes) and results in an action distribution output. The critic baseline uses 3 residual convolution blocks (128 channels, $3 \times 3$ filter size, 2 stride), who's output

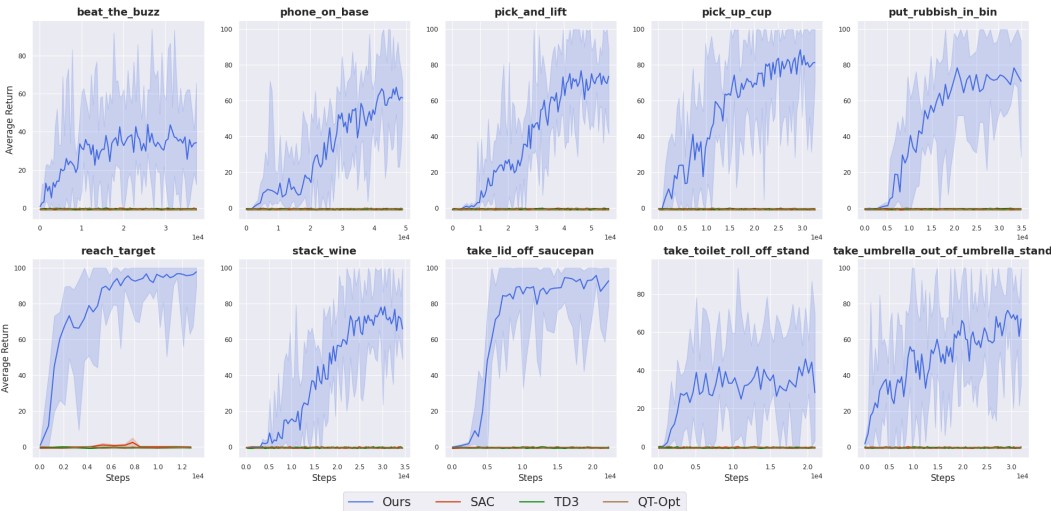

Figure 6: Learning curves for 10 RLBench tasks. Methods include Ours (ARM), SAC (Haarnoja et al., 2018), TD3 (Fujimoto et al., 2018), and QT-Opt (Kalashnikov et al., 2018). ARM uses the 3-stage pipeline (Q-attention, next-best pose, and control agent), while baselines use the 2-stage pipeline (next-best pose and control agent). All methods receive 200 demos which are stored in the replay buffer prior to training. Solid lines represent the average evaluation over 5 seeds, where the shaded regions represent the $min$ and $max$ values across those trials.

feature maps are maxpooled and sent through 2 dense layers (64 nodes) and results in a single Q-value output. All methods use the LeakyReLU activation, layer normalisation in the convolution layers, learning rate of $3 \times 10^{-3}$, soft target update of $\tau = 5^{-4}$, and a reward scaling of 100. Training and exploration were done asynchronously with a single agent (to emulate a real-world robot training scenario) that would continuously load checkpoints every 100 training steps.

The results in Figure 6 show that baseline state-of-the-art methods are unable to accomplish any RLBench tasks, whilst our method is able to accomplish the tasks in small number of environment steps; $5,000$ environment steps equating to about an hour of robot interaction time (meaning *'take_lid_off_saucepan'* being solved in about two hours). We suggest that the reason why our results starkly outperform other state-of-the-art methods is because of two key reasons that go hand-in-hand: (1) Reducing the input dimensionality through Q-attention that immensely reduces the burden on the (often difficult and unstable to train) continuous control algorithm; (2) Combining this with our keyframe selection method that enables the Q-attention network to quickly converge and suggest meaningful points of interest to the next-best pose agent. We wish to stress that perhaps given enough training time some of these baseline methods may eventually start to succeed, however we found no evidence of this.

In Figure 7a, we perform an ablation study to evaluate which of the proposed components contribute the most to the success of our method. To perform this ablation, we chose 2 tasks of varying difficulty: *'take_lid_off_saucepan'* and *'put_rubbish_in_bin'*. The ablation clearly shows that the Q-attention (combined with keyframe selection) is crucial to achieving the tasks, whilst the demo augmentation, confidence-aware critic, and Q regularisation aid in overall stability and increase final performance. When swapping the Q-attention module with a soft attention Xu et al. (2015) module, we found that performance was similar to that of the 'vanilla' baselines. This result is unsurprising, as soft attention is implicitly learned (i.e. without an explicit loss), where as our Q-attention is explicitly learned via (off-policy) Q-learning, and so it can make greater use of the highly-informative keyframes given from the keyframe selection process. Note that we cannot compare to 'traditional' hard attention because it requires on-policy training, as explained in Section 4.1.

Figure 7b shows how robust our method is when varying the number of demonstrations given. The results show that our method performs robustly, even when given 50% fewer demos, however as the task difficulty increase (from *'take_lid_off_saucepan'* to *'put_rubbish_in_bin'*), the harmful effect of having less demonstrations is more severe. Our final set of experiments in Figure 7c shows how our

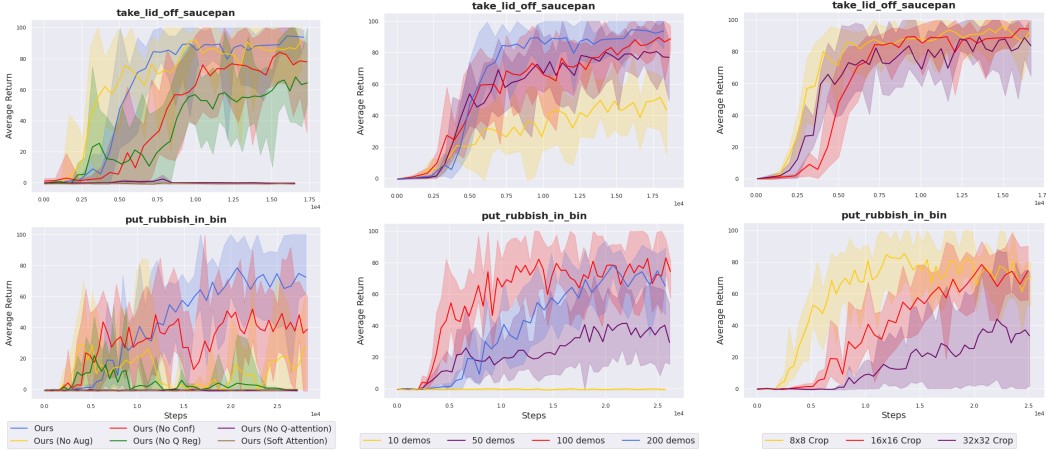

(a) Effect of removing components from our method.

(b) Effect of number of demos on performance.

(c) Effect of crop size on performance.

Figure 7: Ablation study across the easier *'take_lid_off_saucepan'* task and harder *'put_rubbish_in_bin'* task.

method performs across varying crop sizes. As the task difficulty increases, the harmful effect of a larger crop size becomes more prominent; suggesting that one of the key benefits of the Q-attention is to drastically reduce the input size to the next-best pose agent, making the RL optimisation much easier. It's clear that a trade-off must be made between choosing smaller crops to increase training size, and choosing larger crops to incorporate more of the surrounding area. We found that setting the crops at $16 \times 16$ across all tasks performed well.

## 6 CONCLUSION

We have presented our Attention-driven Robotic Manipulation (ARM) algorithm, which is a general manipulation algorithm that can be applied to a range of real-world sparsely-rewarded tasks. We validate our method on 10 RLBench tasks of varying difficulty, and show that many commonly used state-of-the-art methods catastrophically fail. We show that Q-attention (along with the keyframe selection) is key to our success, whilst the confidence-aware critic and demo augmentation contribute to achieving high final performance. Despite our strong experimental results, there are undoubtedly areas of weakness. The control agent (final agent in the pipeline) uses path planning and on-line trajectory generation, which for these tasks are adequate; however, this would need to be replaced with an alternative agent for tasks that have dynamic environments (e.g. moving target objects, moving obstacles, etc) or complex contact dynamics (e.g. peg-in-hole). We look to future work for swapping this with a goal-conditioned reinforcement learning policy, or similar. Another weakness is that we only evaluate on tasks that can be done with the front-facing camera; however we are keen to explore many of the other tasks RLBench has to offer by adapting the method to accommodate multiple camera inputs in future work.

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
