# OpenReview forum: "Attention-driven Robotic Manipulation"
_ICLR.cc/2021/Conference — Reject_

### Official Review · AnonReviewer4 · 2020-10-21
**An interesting solution to training a robotic manipulator using expert examples in sparse-reward  tasks.**

**Rating:** 7
**Confidence:** 4

**Review:**

The paper tackles the problem of training a robotic manipulation model, starting with a set of examples of successful expert trajectories. Training starts with a select set of samples from the expert episodes as the seed of an experience buffer and proceeds to generate new episodes to train on through an RL procedure.

The robotic manipulation model model first applies an attention mechanism in the form of choosing an area of the input images to crop around (though use of a cropping Q function). It then selects the next desired robotic pose from the cropped input using a version of SAC that uses a confidence-augmented Q-function loss and finally applies a motion control mechanism to translate the desired end effector pose into robotic actuation in the simulated setting.

The method makes use of some assumptions regarding what would be the important states and makes use of a selection mechanism to populate an experience buffer. Specifically, it assumes that points in time where the gripper is actuated or velocities approach zero represent necessary steps ("keyframes") in the completion of a task. It then generates examples where the desired robotic action for a state along an example trajectory is the end effector pose of the next keyframe and the desired crop is around the projected location of the desired end-effector pose in the camera image.

Results show that whereas vanilla implementations of SAC, TD3 and QT-Opt with a standard replay buffer can not learn the tasks, the proposed method succeeds. A further ablation study shows the crucialness of the attention-driven cropping step in this setting and the importance of the sample selection strategy and the confidence layer of the SAC Q function.

I find the combination of these applied characteristics of the method to be sensible and powerful:
 - Starting with expert-provided trajectories (and breaking them down into steps / example-poses using the proposed example sampling method)
 - Focusing of the input signal towards the region of interest (bootstrapping with end effector locations) as a better state representation.
 - Using high level control as the action space (next desired pose as opposed to immediate joint actuations)

The method is presented as a general manipulation algorithm. While it does seem to generalize to a rich set of examples in RLBench, it makes use of some properties that are limited to a subset of manipulation tasks. Specifically, tasks where the important activity is limited to a narrow region of the observed field of view and is comprised of steps that can be treated as series of intermediate goals (poses in which velocities are small or the gripper is actuated). It would have been interesting to see whether there are RLBench tasks that should be solvable using a single camera view on which this method fails and to understand the failure reasons.

The methods compared against feel like straw-men. Surely there must be some off-the-shelf methods targeted at learning from expert examples that do not completely fail on these tasks?

There are multiple methods these days that start by transforming the observations into task-oriented state representations (certainly a focus of interest for ICLR). It would have been interesting to see comparisons with alternatives to image crop as a representation (https://arxiv.org/abs/2007.07170 presents one such example).

The paper does not provide code to replicate the experiments. Seeing as it aims to address a known application, this is unfortunate as it would help clarify implementation details and support further research.

Nits:

I found the representation of the point cloud unclear. It seems like it is treated as another RGB from a different view (e.g. cropped to 16x16 pixels?) but if that is true, why call it a point cloud?

The use of the per-pixel Q values (the downstream Q function that has confidence values) is a bit unclear. It might be a no-brainer to someone more familiar with SAC but I found its use to be unclear just from reading this paper. Same is true for the use of 2x Q-functions (referred to as "the double Q trick").

It was unclear to me whether during use, the next iteration of pose selection is performed once the robot has reached the previously selected pose (though the low-level motion control agent) or whether the pose selection occurs at every simulation step regardless of distance from the previously generated pose.

---

> ### Author Response · Authors · 2020-11-11
> **Reviewer 4 Author Response**
>
> We thank the reviewer for the positive review, and are glad that they find the method powerful, and results interesting.  We hope with our detailed clarifications below, that the review will consider strengthening their accept score.
>
> ### *Baselines feel like straw-men. Surely there are methods that do not fail.*
>
> We hope that this paper shows that end-to-end approaches are not suitable as we move from the artificial, densely-rewarded gym environments to these complex, sparsely-rewarded manipulation tasks that are much closer to the reality we face as roboticists. For that reason, we do spend some time discussing why the baselines fail.
> To make these methods succeed, I imagine that if we were to apply techniques such as ‘asymmetric actor-critic’ [1] (where you feed privileged low-dimensional state to the critic) and ‘reset to demonstrations’ [2] (where you periodically reset the simulator/world to a random point in the demonstration and start the episode from there) then we may get some success with vanilla methods. What’s important to realise here is this restricts us to simulation-only training with no way of training in the real world.
> We feel like densely-rewarded baselines such as OpenAI gym and DeepMind control suite can often give us a false sense of progress of continuous RL. As roboticists working at the intersection of ML, we don't have the luxury of having shaped reward in the real world, and so when moving to long horison, sparsely-rewarded tasks that we want to solve in a matter of hours, the cracks begin to show on many state-of-the-art methods.
>
> ### *Task-oriented state representation*
>
> Thank you for pointing out that work. A direct comparison would lead to similar performance to the vanilla baselines as it still has to operate on the full observation input (i.e. it would lack the powerful Q-attention, which is key to the success). However, what could be very interesting for future work would be to combine the GAP method that you linked, with the next-best pose agent; the Q-attention agent would, I imagine, stay roughly the same.
>
> ### *Code*
>
> Yes we plan to release the code for this paper once we have clearance from sponsors.
>
> ### *Representation of point cloud unclear (Figure 2)*
>
> We agree the figure is confusing and will be fixed in the revision. We rotated the visualisation of the organized point cloud to make it clear that it was a point cloud, but in doing so, we can see that this may look like rendering from a different view. To clarity, we do indeed crop an organised point cloud.
>
> ### *Use of per-pixel confidence values is a bit unclear/double q learning trick*
>
> The next-best pose agent is trained with SAC, which is an off-policy actor-critic method. In these sorts of methods, we train the actor (next-best pose agent) by following queues from the critic (the Q-function). Usually, the function Q(s,a) outputs a single value and we use that in the actor update step, but with our confidence-aware critic, we instead output 16x16 values (and confidences), and choose the most confident one instead for the actor. This leads to lower variance when training the actor and thus stabilizes training.
> The double Q-learning trick was made popular by TD3 [3], and essentially means having a duplicate Q-function while training, and using the lowest of the 2 estimates; this helps with the common issue of overconfident Q values while training. This is more of an implementation detail that is common in RL, rather than being core to our method.
>
> ### *Unclear if pose selection is performed when robot reaches the previously selected pose or if it occurs at every simulation step*
>
> The pose selection is chosen when the previously selected pose is reached. We will make this clearer in the new revision. Though in practice, the pose selection could be performed at every Nth time step instead
>
>
> Please let us know if we missed anything, or if you would like further clarification.
>
> [1] Pinto, Lerrel, et al. "Asymmetric actor critic for image-based robot learning." arXiv preprint arXiv:1710.06542 (2017).
>
> [2] Nair, Ashvin, et al. "Overcoming exploration in reinforcement learning with demonstrations." 2018 IEEE International Conference on Robotics and Automation (ICRA). IEEE, 2018.
>
> [3] Fujimoto, Scott, Herke Van Hoof, and David Meger. "Addressing function approximation error in actor-critic methods." arXiv preprint arXiv:1802.09477 (2018).

---

### Official Review · AnonReviewer3 · 2020-10-28
**Interesting approach to attention in robotic manipulation but requires further clarification**

**Rating:** 4
**Confidence:** 4

**Review:**

Summary:

This work focuses on sparse-reward robotic manipulation tasks from image and point cloud inputs, given a few demonstrations, and proposes an algorithm that consists of a Q-attention module and a confidence-aware critic. The Q-attention module is an RL agent, which takes image and point cloud inputs with pixel positions as actions. The image and point cloud observations are then cropped, centered at this pixel position output, and passed down the pipeline to the critic. To make the best use of the demonstrations, the replay buffer is initialized with the demonstrations and further augmented with transitions from demo states, chosen through a keyframe selection process.



Strengths:

- The results demonstrate how, in the difficult setting of learning from sparse rewards and image inputs, only the proposed approach can learn meaningful behavior while the other RL approaches achieve basically zero return. Though, it would be good to include the returns of the expert agent in the plot as well to show the upper bound performance.
- Through an ablation study, it’s clear that the Q-attention agent is important for its ability to reduce the input dimension to the continuous control agent. The demo augmentation process is also quite critical to ensure that the Q-attention and continuous control agents can correctly learn together, since the success of one agent relies on the success of the other agent.
- The paper is overall well-organized and written, and the figures and illustrations are helpful for the reader’s understanding.



Weaknesses:

- It is not completely clear what advantages the Q-attention agent brings over standard attention modules. While a discussion is included, a plot or some other form of quantitative comparison would be more illustrative.
- It’s also not clear why an image crop is the best choice of input for the control agent. For instance, an alternative might be the pixel position itself. This is also highly related to keypoint representations, and there isn’t a discussion of or comparison to them.
- There’s also a key assumption made here that only a fixed region of the image input is relevant to the task at any given timestep. I’m not sure if this is entirely reasonable for environments with different camera viewpoints and different sized objects to manipulate.
- The keyframe selection is based on heuristics: it looks at frames where (1) the gripper state changes as it corresponds to the agent grasping or releasing an object and (2) the agent velocity is near zero as it often corresponds to a new stage of the task.
- The choice of baselines also seems quite arbitrary.



Recommendation:

I am recommending to reject this paper. There are a number of choices in this approach that could be further justified, especially with regards to the Q-attention agent.



Questions:

- Is the pixel position outputted by the Q-attention agent also passed into the continuous control agent? If not, could doing so speed up learning since the cropped image doesn’t provide information about where objects are?

- Instead of a cropped image, is it sufficient to just pass the pixel position, or perhaps the top k pixel positions, given by the Q-attention agent to the continuous control agent? On that note, this is highly related to keypoint representations and it would be good to include a discussion of this in the related work section.

- How well does the Q-attention agent scale to higher-dimensional image inputs? Additionally, how does the crop size affect performance and efficiency of the continuous control agent? That is, with larger objects and a different viewpoint, a much larger crop may be necessary to capture enough context to solve the task.


Minor Comments:

-The labels of Fig. 6 need to be bigger.



Update after Reading Authors' Response:

I appreciate the authors’ thorough response and the new results for different crop sizes. It’s not too surprising that the performance varies with the crop size and that the best crop size depends on the task and environment (among other variables such as camera resolution, camera viewpoint, and object sizes). And so, I share the concern of the other reviewers: it’s unclear whether this approach would be similarly successful in other manipulation settings without prior knowledge of the task outside of RLBench.

---

> ### Author Response · Authors · 2020-11-11
> **Reviewer 3 Author Response**
>
> We thank the reviewer for their positive review. It seems like the majority of the issues can be addressed through clarification and minor edits. We hope with our detailed clarifications below, that the reviewer will increase the rating.
>
> ### *Advantage of Q-attention over other forms of attention*
>
> We want to stress that our contribution is not to try and ‘compete’ with other forms of attention, but instead to highlight that end-to-end methods are not applicable for complex manipulation tasks. But please note that using hard-attention via REINFOCE is not actually possible; REINFOCE is an on-policy algorithm, and is incompatible with the problem setup (would not be possible to include off-policy demonstrations). This is something we failed to mention during initial submission, but in hindsight, this is actually the most important point, and we will certainly include it in a revision. To be clear, we see Q-attention as a type of hard-attention, but with an alternative selection mechanism that can be trained off-policy (through Q-learning) with the output carrying different semantics (score function in the case of REINFOCE hard-attention, and Q-value (expected reward) in the case of Q-attention hard-attention). Since we are allowed 9 pages post rebuttal, we will spend more time discussing these key differences in a revision. Note that we already ran soft-attention baselines in the appendix (which gave similar performance to vanilla baselines). We can add these to the main ablation graph, if the reviewer prefers?
>
> ### *Image crop vs pixel-coordinate input*
>
> The image crop significantly reduces the input dimension to the, much harder, continuous control next-best pose agent. By using a pixel-coordinate, coupled with the full (uncropped) observation, would not have this effect.
>
> ### *Different camera viewpoints and different size objects*
>
> We use the standard, front-facing camera that comes with RLBench. We already discussed in the text the exciting work that can now commence for dealing with multiple cameras from different viewpoints. For example, performing the attention in 3D voxel space (built from combining point cloud information from all cameras). Regarding object size, with any type of hard-attention, a choice must be made about the size of the crop. As you can see by the results, the 16x16 crop was suitable across all tasks. However, we are very happy to run an ablation study with 8x8 crops and 32x32 crops! Please expect this in the upcoming days.
>
> ### *The keyframe selection is based on heuristics*
>
> Could the review clarify why this is a weakness? We actually see this as a major strength; we make simple and sensible assumptions (that require no learning) that hold across all manipulation tasks. We see no motivation to learn these sorts of assumptions for robotics.
>
> ### *Choice of baselines seem quite arbitrary*
>
> Our method is an off-policy RL method, and we compare it to the top 3 state-of-the art off-policy RL methods. It is common to compare RL algorithms against each other in this manner (across multiple seeds). On-policy methods are not applicable because on-policy does not allow the use of demonstrations through a replay buffer.
>
> ### *Pixel-positions passed to the continuous control agent*
>
> Yes they are passed through the pointcloud. Note that it is both the cropped RGB and cropped organized point cloud (that contains x, y, z information) that gets passed to the continuous control agent, and so outputting the pixel location would give no additional information that isn't already included.
>
> ### *Instead of cropping, is it sufficient to pass pixel positions*
>
> It would not be sufficient to only pass the pixel locations as this contains no 3D information, making the next-best pose task impossible. Instead you would have to pass the full 128x128 observations to the continuous control agent, which we have addressed in a previous point (large input dimension, making it very similar to the end-to-end methods that fail).
>
> ### *How does Q-attention scale to higher-dimensional input? How does crop size affect performance?*
>
> You raised a good point about how crop size affects performance! We too are now interested in this. We will add an ablation with a 8x8 crop and a 32x32 crop. Please expect the results soon!
> In terms of the 128x128 input to the Q-attention, we don't imagine that going any smaller than this will be a good idea, as the input point cloud would be far more coarse, and so the accuracy of the next-best pose agent would surely diminish. However, except for an increase in training time and a reduced batchsize, we see no reason why a 256x256 Q-attention input would not work.
>
>
> Please let us know if we missed anything, or if you would like further clarification.

---

### Official Review · AnonReviewer1 · 2020-10-28
**Official Blind Review #1**

**Rating:** 4
**Confidence:** 4

**Review:**

### Summary

Motivated by the fact that RL algorithms are notoriously sample inefficient from pixels and that humans use attention, the authors propose Attention-driven Robotic Manipulation (ARM), which they claim can be applied to several robotics tasks without prior task knowledge. Compared to current methods which fail to train, ARM trains in only a few hours and is successful at the end manipulation tasks. The authors’ main algorithmic improvements with ARM include 1) a Q-attention agent that extracts interesting pixel locations with an explicit attention module, 2) a confidence-aware critic that leads to increased stability of training, and 3) a data augmentation strategy around expert demonstrations.

---

### Strengths

- The paper is well motivated, and the authors recognize a salient problem with most RL systems failing with sparse rewards and being sample inefficient.
- The authors propose several potentially novel contributions, including their Q-attention agent with an explicit attention module, the confidence-aware critic that improves performance, the keyframe discovery strategy, and the demo augmentation method.
- The authors’ demo augmentation strategy is clever in that the required pose transformations are easy to collect and calculate, and this strategy clearly helps performance.
- The authors’ experiments clearly answer the questions they posed around how well this technique learns and how sensitive it is to the number of expert demonstrations. In addition their ablation experiments around which components contribute to the overall pipeline’s success are useful.
- The experimental results show a clear improvement over prior techniques, working with fewer demonstrations and fewer environment steps. The authors’ intuitive explanations for why their method works when others fail are also well-reasoned.

---

### Weaknesses

- The authors claim that their technique can be applied to several tasks without prior task knowledge. This claim only seems weakly supported in the paper. Specifically, to make this claim I would expect to see results across multiple robot platforms. The keyframe selection and demo augmentation strategy does seem specific to the current platform and set of tasks and may not work as well in stochastic environments.
- The authors replace the hand-engineering of reward functions with several hand-engineered modules, as such the theoretical motivation for these modules is less strong. Although I expected to see each module ablated and tested in isolation, these results were lacking in the experiments.
- For the Q-attention agent, the authors claim that a L2 loss prevents overestimation of the Q-values. Because the authors claim that this results was achieved experimentally, such an ablation or analysis should be present in the experiments or supplementary figures.
- The motivation for the Q-attention agent is not made immediately clear. Specifically, the Q-attention agent seems rather similar to performing hard attention with REINFORCE or other policy gradient methods, except here the hard attention is performed with Q-learning. As such, I’m unsure of the actual novelty of this technique.
- The next-best pose agent is particularly confusing since it mentions per-pixel Q-values and per-pixel Bellman losses. This formulation makes sense in the Q-attention, since the action is the pixel being chosen, but makes less sense in the next-best pose agent, since presumably the state is the entire 16x16 window of pixels, and the action is the pose. An ablation on this component’s design would be extremely helpful. It’s not clear to me that the performance gain comes from using 16x16 q-values as opposed to 1x1 q-values, or whether it comes from using confidence values attached to the state action pairs.
- The sparse reward structure being used is missing from the discussion.
- The authors point out that the keyframe selection method likely needs to become more complex with more sophisticated tasks. This again reduces the motivation and impact of the algorithm, if the hand-engineering of the reward function must now be replaced with hand-engineering of the keyframe selection method.
- I would like to see some qualitative analyses of the keyframe selection method and how successful it is. Specifically, what do the sample demonstrations look like, and what percent of frames are keyframes? Intuitively, are there some failure modes of the keyframe selection method where some interesting keyframes are not included?
- I have some confusion around the replay buffers. The pseudocode in algorithm 1 makes it seem that there is a single buffer that stores the final action (pose) taken, but the description claims that the projected pixel coordinates act as optimal actions for the Q-attention agent. Are these projected pixel coordinates also used in a separate replay buffer to warmstart the Q-attention agent?
- It sounds like the proposed methods around data augmentations and discovery strategies are only necessary because the method has multiple agents as opposed to a single end-to-end agent.
- Although none of the compared baselines work on the RLBench tasks, do the authors have some intuition on what would be required to make them successful? Is this due to using a sparse reward structure instead of a dense one? Or not having enough demonstrations? Or not having enough samples to interact with the environment? How does the authors’ method’s performance compare to these baselines in these other regimes? Including these would make the comparison much stronger.
- The ablations present in Figure 6 study whether specific components are necessary (for example whether Q-attention is useful), but not on whether the formulation of that component is the correct formulation (for example, whether Q-attention is better than hard attention). Because the proposed modules are relatively complex and unintuitive, these ablations are critical.
- I would recommend including ablations across all 10 tasks, potentially in the supplementary pages. The current evidence is less convincing that the augmentation strategy or confidence methods are necessary for success, due to the limited number of tasks profiled.

---

### Recommendation

Overall, I vote for rejecting. I think the pipeline proposed by the authors is overly complex, and I am not yet convinced about the motivation for each individual module the authors include. The experimental results are currently lacking in terms of providing this motivation, and missing crucial ablations.

If the authors respond to my comments above with stronger motivations, clearer explanations for how the modules work, and more detailed ablative experiments, their submission will be stronger and many of my concerns would be resolved.

---

### Minor Feedback

- Figure 2 and 3 are unclear since multiple pixels are highlighted, but earlier it is implied that only a single pixel is selected by the Q-attention agent.
- Including videos of demonstrations would be potentially helpful as a qualitative comparison of the behaviors learned between the authors’ proposal and the baselines.
- A qualitative analysis of the failure modes of the authors’ method would also be helpful, but not required.
- Is there a way to visualize the behavior of the next-best pose agent? For instance, how does the pose vary depending on the crop passed in to the agent?
- I don’t understand the reasoning in the appendix (section 7). Specifically, the authors claim that “as in hard-attention style approaches, we are unable to propagate gradients [...] by instead cropping the observation, we alleviate this problem.” The authors’ method still performs non-differentiable cropping, so I don’t see how they are able to propagate gradients. Likewise, cropping observations is commonly done with hard attention, so I don’t see how this method is particularly novel.

---

> ### Author Response · Authors · 2020-11-11
> **Reviewer 1 Author Response (1/2)**
>
> We are surprised by the rating given how strong the strengths were: the review points out that the paper is well motivated, has 4 main contributions, includes strong ablation experiments, and good discussion on why our method beats state-of-the art. It seems like most of the issues can be resolved with some clarification on our side. We hope with our detailed clarifications below, that the reviewer will increase the rating.
>
> ### *Claim that technique can be applied to several tasks without prior knowledge*
>
> To be clear, we mean robot manipulation tasks, and we believe the broad range of tasks that are present in RLBench support this claim. The keyframe selection is robot arm agnostic because they operate in the end-effector space (e.g. gripper open/close, sudden stops, etc), and not the joint space.
>
> ### *Replace hand-engineered reward with several hand-engineered modules.*
>
> Could the reviewer please clarify what part of the method they believe is hand-engineered? The method is the same across all tasks, with no per-task hand-engineering. Reward shaping on the other hand requires per-task hand-engineering. Also, we are a little confused by the last sentence, as we do ablate each of the modules, and in fact the reviewer highlights this as one of the strengths. Perhaps we are misunderstanding the reviewers point here?
>
> ### *L2 loss preventing overestimation of the Q-values*
>
> We are happy to provide an additional ablation with the L2 loss omitted. Please expect the results in the next revision.
>
> ### *Q-attention vs hard attention with REINFORCE / ablation*
>
> We want to stress that our contribution is not to try and ‘compete’ with other forms of attention, but instead to highlight that end-to-end methods are not applicable for complex manipulation tasks. But please note that using hard-attention via REINFOCE is not actually possible; REINFOCE is an on-policy algorithm, and is incompatible with the problem setup (would not be possible to include off-policy demonstrations). This is something we failed to mention during initial submission, but in hindsight, it is actually the most important point, and we will certainly include it in the revision. To be clear, we see Q-attention as a type of hard-attention, but with an alternative selection mechanism that can be trained off-policy (through Q-learning) with the output carrying different semantics (score function in the case of REINFORCE hard-attention, and Q-value (expected reward) in the case of Q-attention). Since we are allowed 9 pages post rebuttal, we will spend more time discussing these key differences in a revision. Note that we already ran soft-attention baselines in the appendix (which gave similar performance to vanilla baselines). We can add these to the main ablation graph, if the reviewer prefers?
>
> ### *Clarification on the next-best pose agent*
>
> The per-pixel confidence can be seen as a sort of per-pixel voting in order to make sure that we use the most confident value to update the actor. This is particularly important because continuous actor-critic algorithms are much more unstable to train than discrete Q-learning algorithms (e.g. the Q-attention). It would be impractical to use the 16x16 values without the confidence as we would have no way of choosing which of the 16x16 values to use when updating the actor network. If we were to pick the highest of the 16x16 values, then we have no guarantee that this value is meaningful (i.e. it could have no idea and output an arbitrary high value, which would lead to an incorrect signal to the actor)
>
> ### *Sparse reward structure*
>
> We use the reward as defined in the RLBench paper, which is +1 on task completion, 0 elsewhere. We will add a sentence in the text to clarify this.
>
> ### *Keyframe selection likely needs to be more complex*
>
> Indeed we mention this in the paper to self-critique ourselves and to inspire research in other forms of keypoint detection. However, to be absolutely clear, at no point did we find a task that did not adhere to the selection criteria that we laid out. I believe the term ‘hand-engineered’ here seems unsuitable given that these are not per-task tuned (unlike reward-shaping) but simply assumptions that we make about when interesting points are likely to happen. Please let us know if you need further clarification, as this is a very important point.
>
> ### *Qualitative analyses of keyframe selection. Any modes of failure.*
>
> We could indeed show a figure (similar to our attention across time in Figure 3) where we show the observation at each keypoint across a few tasks. Now that we have 1 additional page, we can certainly show this if it would aid reliability. Regarding failure modes, we did not see any situations where interesting keyframes were not included.

---

> > ### Author Response · Authors · 2020-11-11
> > **Reviewer 1 Author Response (2/2)**
> >
> > ### *Confusion around replay buffers*
> >
> > There is indeed a single replay buffer that stores (in addition to the usual state, reward, etc) both the pixel coordinate action and the next-best pose action; the former being used to update the Q-attention, and the latter for the next-best pose agent. We hope Algorithm 1 makes this clear, but we will also explicitly state this in the text to make it clearer.
> >
> > ### *Data augmentation/discovery strategy necessary because the method has multiple agents vs end-to-end.*
> >
> > As we showed in the ablation: the data augmentation is not necessary, but does increase performance as the tasks get more difficult. The keypoint discovery is the necessary part that helps train the Q-attention. As you can see in the results, all this is necessary because end-to-end methods do not work on these complex, sparsely-rewarded manipulation tasks.
> >
> > ### *Intuition of why baselines fail*
> >
> > We hope that this paper shows that end-to-end approaches are not suitable as we move from the artificial, densely-rewarded gym environments to these complex, sparsely-rewarded manipulation tasks that are much closer to the reality we face as roboticists. For that reason, we do spend some time discussing why the baselines fail (which you stated is one of our strengths). To answer your question directly, to make them successful, we imagine that if we were to apply techniques such as ‘asymmetric actor-critic’ [1] (where you feed privileged low-dimensional state to the critic) and ‘reset to demonstrations’ [2] (where you periodically reset the simulator/world to a random point in the demonstration and start the episode from there) then we may get some success with vanilla methods. What’s important to realise here is this restricts us to simulation-only training with no way of training in the real world.
> >
> > ### *Ablations across all 10 tasks*
> >
> > We are happy to do this, but this will take much longer to complete than the 2 week rebuttal period. We urge the reviewer to consider the amount of compute needed to run all ablations, across 5 seeds, across all 10 tasks. The results you see have already taken several months to generate given our broad range of off-policy baselines. This is why we chose to perform the ablation on 2 tasks that are representative of the others (e.g. one easier task and one harder task).
> >
> > ### *Fig 2 and 3 unclear because multiple pixels highlighted*
> >
> > Multiple pixels are highlighted because they are showing the Q-value of each pixel. We select the pixel with the highest value for the cropping. You can think of these as attention maps.
> >
> > ### *Videos of demonstrations*
> >
> > Videos of demonstrations can be seen from the RLBench site (https://sites.google.com/view/rlbench). We can, however, copile a new video showing the demos of the tasks that were included in this paper to aid readers.
> >
> > Please let us know if we missed anything, or if you would like further clarification.
> >
> > [1] Pinto, Lerrel, et al. "Asymmetric actor critic for image-based robot learning." arXiv preprint arXiv:1710.06542 (2017).
> > [2] Nair, Ashvin, et al. "Overcoming exploration in reinforcement learning with demonstrations." 2018 IEEE International Conference on Robotics and Automation (ICRA). IEEE, 2018.

---

### Author Response · Authors · 2020-11-16
**Revision Uploaded**

The additional experiments that we stated in the individual responses are now done, and we have uploaded a new revision. Here are the changes:

- (R1) Added ablation in Fig 7a on L2 reg (we call this Q regularisation). As stated in the main text, the lack of L2 loss on the per-pixel output leads to stability issues.
- (R1) Clarified the sparse reward (+1 on task completion, 0 otherwise).
- (R1) A new figure (Fig 5) showing the result of keyframe selection on a handful of tasks.
- (R1) We have included a video showing examples of demonstrations in the supplementary material.
- (R1/R3) We have improved the distinction of soft-attention, hard-attention, and Q-attention (see end of Section 4.1). We have also included the plot of an additional soft-attention ablation in Fig 7a. As explained in our individual responses, it is not possible to compare to ‘traditional’ hard attention because of its on-policy nature (we also clarified this in the paper in the results section).
- (R3) Added results on additional crop sizes; Figure 7(c) now has a plot for crop sizes of 8x8, 16x16, and 32x32.
- (R3) Labels on Fig 7 are now bigger
- (R4) Organized point cloud visualisation in Fig 2 improved.
- (R4) Added clarification that the next-best pose agent is run when the agent reaches the previously selected pose.

---

### Decision · Program_Chairs · 2021-01-07
**Final Decision**

**Decision:**

Reject

**Comment:**

This paper proposes  a general manipulation algorithm for tasks that have sparse rewards. The algorithm uses a Q-attention to extract interesting pixel locations with an explicit attention module. A data augmentation method is also proposed to generalize expert demonstrations.
While the proposed method and experiments seem interesting, two out of the three reviewers had several issues regarding the clarity of the paper. The main issue is that a better ablation study needs to be performed to assess the proposed system. For instance, the reviewers question the advantages the Q-attention agent brings over a standard attention module. The authors provided detailed explanations and a video showing examples of the expert demonstrations. It is not clear however if the clarity issues are completely resolved.